



# MIMiX: A Multipurpose In-situ Microreactor system for X-ray microspectroscopy to mimic atmospheric aerosol processing

Jan-David Förster[1,2], Christian Gurk[3], Mark Lamneck[3], Haijie Tong[1], Florian Ditas[1], Sarah S. Steimer[4], Peter A. Alpert[5], Markus Ammann[5], Jörg Raabe[6], Markus Weigand[7], Benjamin Watts[6], Ulrich Pöschl[1], Meinrat O. Andreae[2,8], and Christopher Pöhlker[1,2]

[1]Multiphase Chemistry Department, Max Planck Institute for Chemistry, 55128 Mainz, Germany
[2]Biogeochemistry Department, Max Planck Institute for Chemistry, 55128 Mainz, Germany
[3]Instrument Development Group, Max Planck Institute for Chemistry, 55128 Mainz, Germany
[4]Department of Environmental Sciences, University of Basel, 4001 Basel, Switzerland
[5]Laboratory of Environmental Chemistry, Paul Scherrer Institute, 5232 Villigen PSI, Switzerland
[6]Laboratory Condensed Matter Physics, Paul Scherrer Institute, 5232 Villigen PSI, Switzerland
[7]Institute for Nanospectroscopy, Helmholtz-Zentrum Berlin für Materialien und Energie GmbH, 12489 Berlin, Germany
[8]Scripps Institution of Oceanography, University of California San Diego, La Jolla, CA 92037, USA

**Correspondence:** Jan-David Förster (jd.forster@mpic.de) & Christopher Pöhlker (c.pohlker@mpic.de)

**Abstract.** The dynamic processing of aerosols in the atmosphere is difficult to mimic under laboratory conditions, particularly on a single particle level with high spatial and chemical resolution. Our new microreactor system for X-ray microscopy facilitates observations under *in-situ* conditions and extends the accessible parameter ranges of previously reported setups to very high humidities and low temperatures. With the parameter margins for pressure (180-1000 hPa), temperature ($-23\,°C$ to room temperature), and relative humidity (∼ 0 % to above 98 %), a wide range of tropospheric conditions is covered. Unique features are the mobile design and compact size that make the instrument applicable to different synchrotron facilities. Successful first experiments were conducted at two X-ray microscopes, i) MAXYMUS, located at beamline UE46 synchrotron BESSY II, Berlin, Germany, and ii) PolLux, located at beamline X07DA of the Swiss Light Source in the Paul Scherrer Institute, Villigen, Switzerland. Here we present the design and analytical scope of the system, along with first results from hydration/dehydration experiments on ammonium sulfate and potassium sulfate particles and the observation of water ice at low temperature and high relative humidity in a secondary organic aerosol particle from isoprene oxidation.





# 1 Introduction

Aerosol particles play crucial roles in various atmospheric processes and the Earth's climate system (e.g., Pöschl, 2005; Andreae and Rosenfeld, 2008; Kolb and Worsnop, 2012; IPCC, 2013). Precise knowledge of their physical and chemical properties on a single particle level (i.e., mixing state, hygroscopicity, viscosity, occurrence of phase separation) is needed to correctly evaluate the aerosols' atmospheric influence. Accordingly, a focal point of current aerosol research is to retrace the dynamic life cycle of aerosol particles in the atmosphere upon cloud processing, chemical aging, and the associated multiphase processes (e.g., Mikhailov et al., 2009; Koop et al., 2011; Shiraiwa et al., 2013; Pöschl and Shiraiwa, 2015).

Scanning transmission X-ray microscopy with near-edge X-ray absorption fine structure analysis (STXM-NEXAFS) in the soft X-ray regime (270-2000 eV) has become a widely used and powerful technique to resolve the micromorphology and chemistry of laboratory and ambient aerosol particles on submicron scales (e.g., Moffet et al., 2011; Shakya et al., 2013; O'Brien et al., 2015). However, most analyses of this kind were conducted on dried particles impacted on sampling substrates, representing a strongly altered state in relation to the particles' microphysical conditions in the atmosphere. Accordingly, some studies on laboratory generated standard aerosols have combined STXM-NEXAFS analyses with observations under more authentic atmospheric conditions, such as varying relative humidity (RH) levels (e.g., Ghorai and Tivanski, 2010; Zelenay et al., 2011a, b, c; Steimer et al., 2014). Ambient aerosol particles, which we investigated with STXM under varying RH conditions, showed remarkable changes in microstructure and phase state as a function of RH (Pöhlker et al., 2014). While these initial studies have provided interesting insights into the dynamic life cycle of aerosol particles in the atmosphere, results of this kind - particularly on collected ambient particles - have remained sparse due to technical challenges in reliably controlling the temperature ($T$), pressure ($p$), and RH over the sample throughout the course of the already challenging STXM experiments. As an example, the comparatively simple experimental setup in Pöhlker et al. (2014) was inherently limited by low $p$ conditions, RH < 87 %, and unregulated $T$.

Here, we present the development of a gas flow system coupled with a microreactor as an accessory for STXM instruments for *in-situ* studies of particles in a controlled gas-phase environment, we emphasize its analytical capabilities and show initial results. The instrument's design and construction was inspired by previous developments of environmental chambers for X-ray microscopes, namely by Drake et al. (2004), de Smit et al. (2008), Huthwelker et al. (2010), and Kelly et al. (2013). The microreactor system has been developed according to the following requirements:

- Compactness and portability:
  facilitating compatibility of the system with different STXM instruments and application at different synchrotron sites

- Minimal optical path length:
  accounting for short focal lengths in STXM optics, e.g., to allow measurements at the carbon (C) $K$-edge and at even lower energies





- – Maximum sample compatibility:

  suitable for standard silicon nitride membrane windows ($500 \times 500\,\mu m^2$) operated at up to $1000\,hPa$ pressure difference between the inside of the microreactor and the surrounding STXM enclosure, quick and safe sample (un-)mounting

- – Reliable and stable parameter control:

  environmental parameters $p$, $T$, and RH tunable over a wide value range; particularly, humidity control in high RH regime (i.e., $80\,\%$ RH up to saturation) and control over $T$ below $0°C$ for kinetic studies and freezing experiments

- – Extension options:

  interfaces to the gas supply circuit for the introduction of reactive atmospheres to study particle/gas phase reactions (e.g., ozonolysis)

## 2 Technical Description

### 2.1 Control System Design

The control system is of compact size: The gas mixing and cooling circuits, along with the power converters and electronics are integrated into a 19-inch enclosure with a height of 4 rack units (total dimensions: $37 \times 48 \times 17.5\,cm^3$). The relevant parts of these circuits and the positions of the environmental sensors (BME280 by Bosch Sensortec (2019)) therein, which were used throughout the system to trace changes of the $p$, $T$, and RH values are shown in Fig. 1. The only external supplies needed, besides mains voltage, are a vacuum pump and a source of pressurized process gas, such as nitrogen or synthetic air, but preferably helium to achieve the best signal-to-noise ratio across all reachable absorption edges. However, provision was made for operation under reactive (e.g., ozone-enriched) atmospheres by using external gas supplies, indicated by the orange dashed line in Fig. 1, which can be attached to the system via designated ports.

The tasks of control and data acquisition are performed by the so called 'VBUS system', which has been developed at the Max Planck Institute for Chemistry (MPIC). This miniature measurement system consists of microcontroller-based electronic modules and a flexible software environment including scripts and a graphical user interface (GUI). An example GUI screenshot is provided in the supplementary Fig. S1. The gas humidification system is similar to the one used by Huthwelker et al. (2010) and mixes wet and dry gas flows to provide a process gas of desired humidity. Upstream, two Bronkhorst® IQ⁺FLOW® IQFD-200C mass flow controllers (MFC) assure that the combined flows equal $20\,mL\,min^{-1}$. By running one of the gas streams through a Permapure Nafion humidifier (MH-110-24F-4, $60\,cm$ length), the RH directly after the humidifier can be increased to up to $70\,\%$ RH at ambient system temperature and should not exceed this value to avoid condensation inside the control system. The RH values are measured, along with $p$ and $T$, at three locations within the circuit by Bosch BME280 environmental sensors (Bosch Sensortec, 2019), indicated by circles in Fig. 1, labeled with "p, T, RH" and named accordingly: S1) is located directly downstream of the humidifier. Its RH value is taken as the control variable, which is continuously checked at a rate of $4\,Hz$ against a desired set point. The RH reading typically fluctuates by about $\pm 0.05\,\%$ RH, which can be taken as the relative accuracy of the software-implemented proportional-integral (PI) controller that steers the MFCs' flows; S2) measures $p$, $T$, and


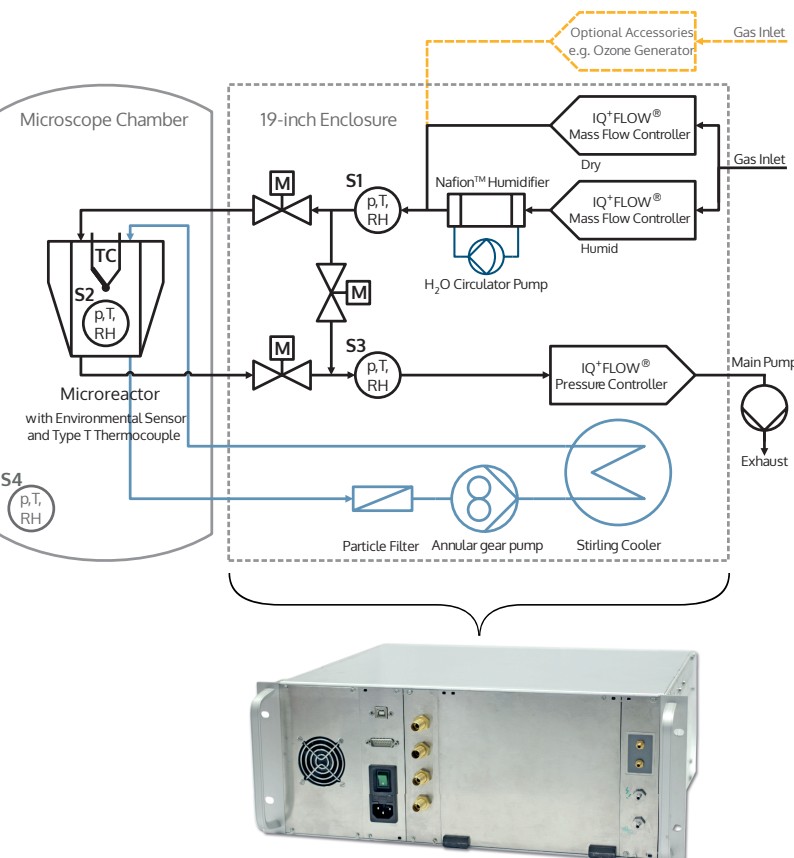

**Figure 1.** Gas Flow and cooling system schematics of the entire system. Left: X-ray microscope chamber with sketched microreactor. Refer to Fig. 2 for a detailed view of the microreactor. Right: 19-inch enclosure that includes the control system. S1-S4: p, T, RH environmental sensors (Bosch BME280); M: Solenoid operated shut off valves (Bronkhorst® EV-02-NC-V); TC: Thermocouple (Omega™ 5TC-TT-TI-4)

RH values inside the microreactor close to the sample; S3) sits symmetrical to S1 in the return flow. This sensor is particularly useful to detect losses due to leaks or condensation inside the flow system. S1 and S3 are interfaced via the Inter-Integrated Circuit ($I^2C$) bus, S2 is mounted onto a separate printed circuit board (PCB) with Serial Peripheral Interface (SPI) capabilities (compare with Fig. 2).

Stable temperature, $T$, is critical for accurate RH control. Therefore, our design includes a closed circuit active cooling
system, which combines a Twinbird Corp. SC-UB04 Free Piston Stirling Cooler with a HNP Mikrosysteme mzr®-2921 micro annular gear pump. As a coolant the hydrofluoroether $C_4F_9OC_2H_5$ was used, commercially available as 3M™ Novec™ 7200 Engineered Fluid. By controlling the pump speed between 0.3 and 18 mL min$^{-1}$ via a second software PI-regulator that takes the $T$ value from inside the microreactor as the control variable, temperatures between the ambient $T$ and $-23\,°C$ can be set. The $T$ control system stability, the sensor characteristics, and possible applications for the cooling capabilities will be detailed
in section 3.


The system's pressure, $p$, can be controlled between ∼ 180 hPa and 1000 hPa, by using a Bronkhorst® IQ+FLOW® IQPD-700C Back Pressure Controller downstream of the flow circuit. This controller is also active during the evacuation of the microscope chamber to keep the differential pressure between the microreactor and the surrounding, monitored by S2 and S4 (refer to Fig. 1), respectively, at a minimum, if not desired otherwise. Not shown in the flow scheme in Fig. 1 are ports for

venting the STXM vacuum chamber for the reverse case. Three Bronkhorst® solenoid operated shut off valves (EV-02-NC-V) can be used to bypass the microreactor, e.g., for leak testing. If not stated otherwise, the gas and fluid system components, fittings, filters and connectors used in this system were purchased from Swagelok. The essential parts of the flow circuit are surface mounted to an aluminum circuit board, which interconnects them via internal channel structures. A detailed view is given in the supplement in Fig. S2.

## 2.2 Microreactor Design

The microreactor, displayed in Fig. 2, serves multiple purposes: It holds the sample in place at a defined $T$, and exposes it to a process gas previously mixed inside the control system. All custom designed parts were computer-numerical-control (CNC)-machined in the mechanical workshop of the MPIC. The individual components were conceptualized in Autodesk® Inventor 2014 as computer-aided design (CAD) models, which were used for the renderings in Fig. 2 and 3.

The fluidic connections between the front panel of the control box and the microreactor were realized via four Upchurch Scientific (IDEX Health & Science) PEEK tubings (1.6 mm outer diameter x 1.0 mm inner diameter x 1.5 m length) for the gas and coolant flows, guided through a custom-made vacuum feed-through. A 4-way microfluidic connector (Dolomite Ltd., Part number 3000024) then seals the microreactor body (Fig. 2) from the interior of the STXM chamber. The reactor body itself consists of two CNC-machined brass metal parts with internal channel structures, glued together with LOCTITE® EA 9497

two-component epoxy adhesive.

The microreactor features a quick-change mechanism via twist-lockable sample mounting disks, which are 1.5 mm thick, manufactured from an aluminum alloy. The usage of these disks allow a tension-free sample exchange and minimizes the number of separate parts, such as screws. Standard silicon nitride membrane windows ($500{\times}500{\times}0.1\,\mu m^3$ membranes with $5{\times}5{\times}0.2\,mm^3$ outer silicon frame dimensions) act as the sample substrate and are mounted to these plates, which simulta-

neously act as the microreactor's front cover. For exchanging the sample, the mounting plate needs to be pressed downwards against the brass metal body to compress the underlying O-ring seal. A 45° rotation then locks or unlocks the sample, respectively. For instance, this can be done with sharp tweezers. After mounting the sample, the expanding O-ring in its groove presses the sample holder against its counterpart, the front steel plate, which keeps the sample at a defined position and well-sealed. Sectional views in Fig. 3b and c illustrate how the parts are assembled. A rendered image sequence, composed to a

video clip by using the video editing software DaVinci Resolve 16, shows the assembly of the microreactor, emphasizes its internal structures, the locations of the sensors, and the O-ring seals, and visualizes the sample mounting process. It is provided as a Video supplement below.

With a typical focal length of 1.2 mm at 280 eV, the space between the focal plane and the zone plate (compare Fig. 3a), with the order sorting aperture (OSA) in between, is very limited. The OSA is located approximately 350 μm away from the





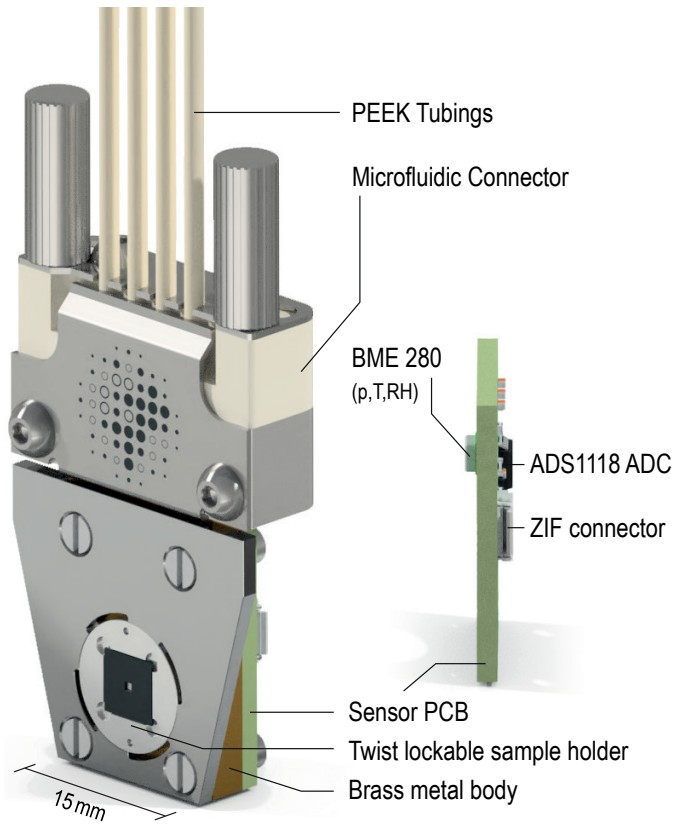

**Figure 2.** Rendered views of the microreactor assembly and the sensor PCB with the most important parts labeled. The 4-way microfluidic connector (Dolomite Ltd.) connects to the internal channel structure in the brass metal body of the microreactor via a compression seal at the top face. The signals from the BME280 ($p$,$T$,RH) sensor (S2 in Fig. 1) and the ADS1118 16 bit analog-to-digital converter (ADC) with internal $T$ reference sensor for the thermocouple readout are available through an Serial Peripheral Interface (SPI) at the 6-pin ZIF connector.

sample plane to prevent unfocused X-rays and stray light from passing through the sample and from entering the detector. Subtracting the silicon frame thickness of the front window leaves just 150 µm between the OSA and the microreactor. We therefore avoided a sample holder design that adds more material to this side of the microreactor. For a gas-tight seal, the silicon nitride windows must be glued into the sample holder disks. We either used IMI 7031, also known as GE-varnish, and let it cure at room temperature or Apiezon Wax W, which melts at about 100 °C and therefore is only suitable for empty

windows prior to sampling or temperature insensitive samples.

     Behind the front window, the process gas flows in a 300 µm wide gap, as emphasized in Fig. 3b. This specific gap width was chosen for maintaining sufficient X-ray transparency while having enough room for a thermocouple (Omega™ 5TC-TT-TI-40), which is in loose contact with the silicon frame of the front window to provide a $T$ reading from as close to the sample as possible. The pressurized gap is closed towards the detector side by a second window with a circular frame. The detector-

side window, 3 mm in diameter with 0.2 mm frame thickness, was glued to the tapered aluminum insert, which surrounds the





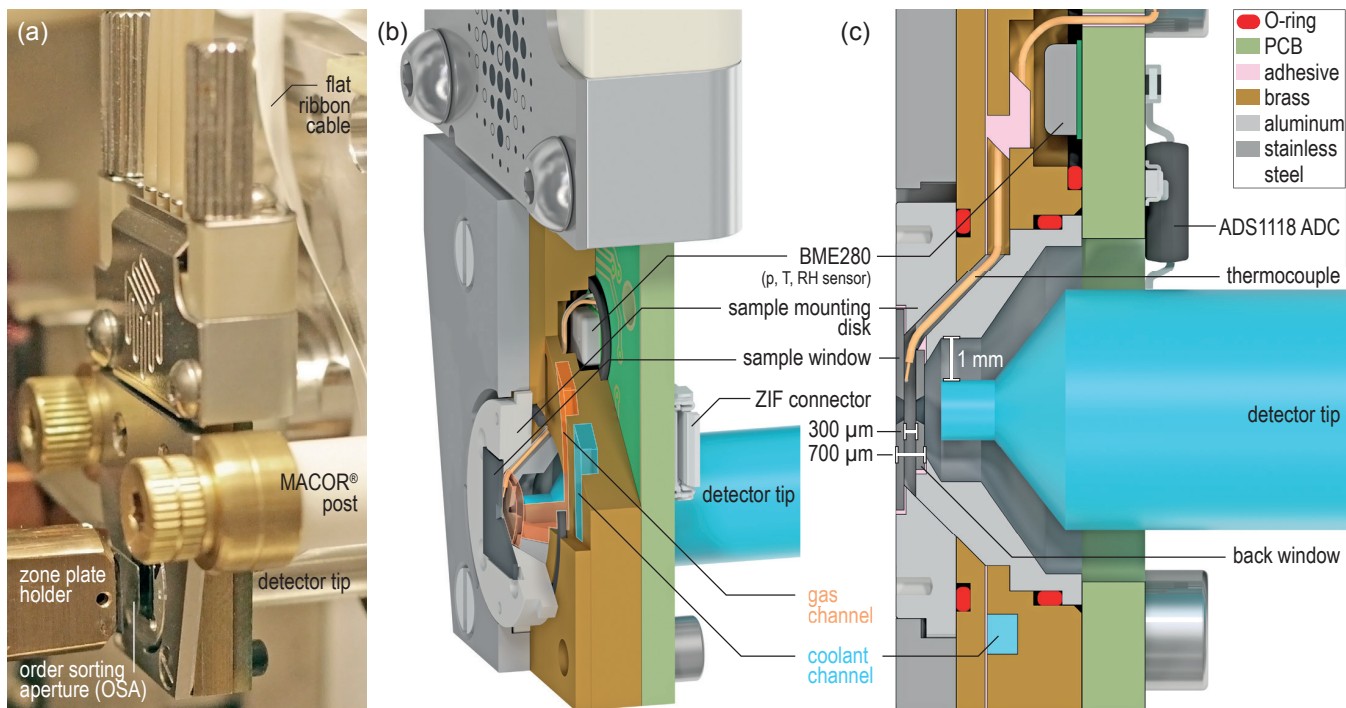

**Figure 3.** a) The microchamber adapted to the MAXYMUS instrument on a customized holder, thermally insulated on MACOR® posts. This photo emphasizes the spatial restrictions due to the integral parts of the microscope (zone plate, OSA, detector). b) Sectional view through the microchamber, revealing the internal channel structures, the positions of O-rings, the sensor cavity and the groove into which the thermocouple wires are firmly glued. c) Cross-sectional view at a larger scale, emphasizing the physical thickness of the microreactor along the optical axis (700 μm), given by the thicknesses of the silicon nitride window frames, 200 μm on each side, and the gap width of 300 μm, in which the gas stream flows. The microreactor can be freely driven by one 1 mm in every direction in the focal plane without colliding with the detector.

detector tip, with the same adhesive as the front window. Consequently, the physical thickness of the microchamber along the optical axis only measures 700 μm (compare Fig. 3c). With a back-window membrane size of $1000{\times}1000{\times}0.05\,\mu m^3$, unobstructed views of the front membrane at the carbon $K$-edge with differential pressures to up to 500 hPa are possible. Usually membranes with $500{\times}500{\times}0.05\,\mu m^3$ were used, as vignetting in the outer regions of the sample window was less

important to us compared to increased burst resistance at atmospheric $p$. Please note that the inward facing of the silicon nitride windows is crucial for assuring the best $p$ resistance, i.e. a safe operation at 1000 hPa differential pressure. An outward facing front window as shown in Huthwelker et al. (2010) is prone to delamination of the silicon nitride film and subsequent burst of the membrane.

     The rearmost part of the microreactor is a PCB, which is screwed to the brass metal body. It includes read-out electronics

(ADS1118 ADC) for the thermocouple, an electrical connector, as well as the environmental sensor S2, which reaches into the gas stream through an O-ring-sealed recess in the metal body (Fig. 3B). A total number of three O-rings seal the microreactor





from the microscope. The O-rings were slightly lubricated with Apiezon N Cryogenic High Vacuum Grease to assure a tight seal and a smooth rotation of the sample locking mechanism. We could not detect any carbonaceous contaminants from a potential outgassing of the O-rings, from the lubricant or from the glued components. The microreactor gets constantly flushed
with fresh process gas at comparably high flow rates.

## 3  Performance evaluation

### 3.1  Parameter control and stability

A wide range of atmospheric conditions present in the troposphere can be reproduced by the microreactor system. More precisely, the system uses pressures ranging from 180 hPa to 1000 hPa and has cooling capabilities for highly stable $T$ controlling
between room temperature and approximately -23 °C, depending on the insulation quality in the individual setup. All parameters are actively regulated via feedback control systems. The (controllable) working range for the RH spans from dry conditions to above 98% RH, depending on the residual moisture of the process gas, and the temperature difference between the humidifier and the microreactor body. Due to heat dissipation by electronic components, a control system $T$ of 28-29 °C, measured inside the aluminum circuit board, is typical. A 65 % RH at sensor S1 directly after the humidifier, therefore translates into saturation
conditions at a microreactor $T$ of 22 °C. Steep $p$ and RH gradients (e.g., from dry conditions to above 80 % RH within less than a minute) can be achieved in a reliable and reproducible manner. Besides a stepwise increase of the humidity, it is possible to run preprogrammed ramps or periodically repeated hydration/dehydration cycles to mimic dynamic atmospheric processes.

In Fig. 4a the RH trend upon a stepwise increase of the setpoint over a time period of ~ 40 minutes is shown. Noticeable is the discrepancy between the RH measured at the exit of the humidifier (sensor S1) and inside the microreactor body (sensor
S2) as a result of the temperature difference and a slowed response of the S2 reading due to diffusion inside the gas stream and the wetting of exposed surfaces. In the particular example shown here, the microreactor was held at 10 °C and the ambient system $T$ varied between 28-29 °C. This experiment can be seen as a typical example for sensing the deliquescence point of ammonium sulfate (AS) at 82 % RH at 10 °C, as reported by Tang and Munkelwitz (1993).

A $T$ control with the least possible fluctuation margin is crucial, especially at high RH close to saturation conditions to
avoid undesired water condensation inside the microreactor. Therefore, the $T$ control system was designed for high-precision regulation and not for high cooling (~ 2 °C min⁻¹) or warming rates. In Fig. 4b, the high stability of the regulator set to a target $T$ of 10 °C is shown over a 4-hour time span.

The $T$, as well as the $RH$ regulation is based on software PI-controllers running as scripts inside the 'VBUS' software environment. In a test case, $T$ as low as -23 °C could be reached inside the microreactor. For details, refer to Fig. S1 in the
supplement. Besides kinetic studies for the measurement of reaction rates, with this capability, freezing experiments, and the study of diffusion-controlled reactions at a single particle level, come into reach.

Note that the thermocouple in principle is redundant for the temperature measurement inside the microreactor body, as the BME280 sensor (compare Fig. 2) usually gives the same temperature readings, but with a higher resolution. Any heat transfer mechanisms besides radiant heat transfer are minimized within the microreactor. Convective heat transfer is suppressed by the

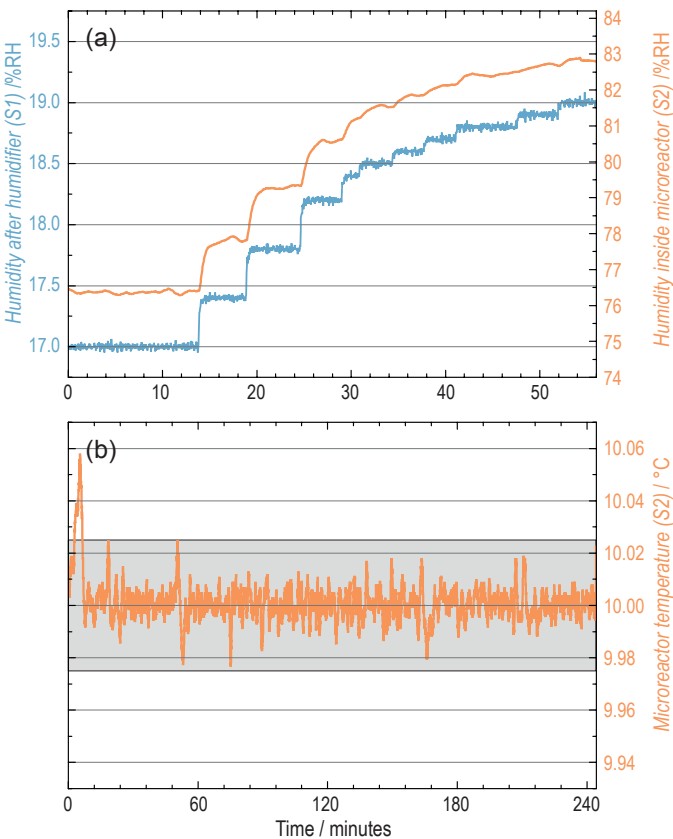

**Figure 4.** a) Response characteristic of the humidifier at different setpoints. blue: humidity measured at S1 (control variable); orange: humidity measured at S2 (compare with Fig. 1) at an ambient temperature between 28-29 °C and a microreactor $T$ of 10.0 °C. b) Microreactor $T$ stability over ~ 240 min at a setpoint of 10.0 °C. The gray shading emphasizes a $\pm 0.025$°C margin.

surrounding vacuum and the microreactor body is clamped to the microscope's stage, thermally well insulated by MACOR® ceramic posts, which can be seen in Fig. 3a. However, for very low temperatures, a discrepancy of a few tenths of degrees at -23 °C between both sensors was present, increasing when temperatures are lowered. We attribute this to the radiant heat transfer between the OSA and the sample window.

### 3.2  Sensor calibration and initial tests

The BME280 environmental sensors used in this setup were chosen because of their very small dimensions of $2.5 \times 2.5 \times 0.93$ mm³ (Bosch Sensortec, 2019). The BME280 sensors' and the thermocouple's temperature readings were calibrated in a cooling bath between -10°C and 27°C against a reference thermometer (Fluke 2180A, Fluke Deutschland GmbH with 0.01°C resolution and a minimum uncertainty of $\pm 0.08$°C. The humidity readings were calibrated using the deliquescence relative humidities (DRH) of salt standards documented in the literature (e.g. NaCl, $(NH_4)_2SO_4$, $K_2SO_4$ with DRH values





of 75.3±0.1 % RH (Tang and Munkelwitz, 1993), 80 % RH (Tang and Munkelwitz, 1994), and 97.6±0.6 % RH (Greenspan, 1977), respectively). For test measurements, the setup was successfully adapted to two STXM instruments, i) MAXYMUS, located at beamline UE46-PGM-2 of the synchrotron BESSY II, Berlin, Germany (Follath et al., 2010; Nolle et al., 2011; Weigand, 2014), and ii) PolLux, located at beamline X07DA of the Swiss Light Source in the Paul Scherrer Institute, Villigen, Switzerland (Flechsig et al., 2007; Raabe et al., 2008; Frommherz et al., 2010).

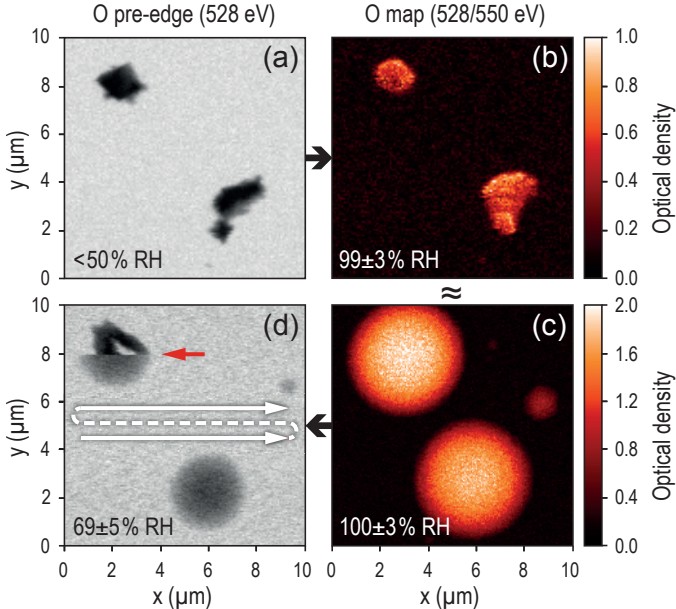

**Figure 5.** Image sequence showing the hydration (a→c) and dehydration (c→d) of $K_2SO_4$. The RH-values here represent raw sensor data. Panel (a) represents dry conditions, (b) shows the situation shortly before deliquescence. An aqueous shell has already formed around the particles and crystal edges are rounded off. In panel (c) full deliquescence has already occurred. The deliquescence point was reached between 99-100 % RH at 21 °C. This is in good agreement with the literature value of 97.6±0.6 % RH (Greenspan, 1977) taking into account the sensor accuracy. Efflorescence was observed during the scan of the image in panel d) with a sharp transition (red arrow) at 69±5 %RH, in contrast to the literature value 60 % RH, reported by (Freney et al., 2009). Note that white arrows indicate the direction of the raster scan pattern during the measurements. The images in the left column (a & d) represent raw x-ray absorption data at 528 eV and in the right column (b & c) optical density (OD) maps are shown to emphasize the oxygen distribution. The maps were generated with the Multivariate ANalysis Tool for Spectromicroscopy software (MANTiS v.3.0.0.1) (Lerotic et al., 2004, 2005, 2014) in commit version #5847171 (Lerotic et al., 2019).

A hydration/dehydration experiment with $K_2SO_4$ particles was measured at PolLux as a first proof of performance application for the microreactor system. The results shown in Fig. 5 illustrate the operation of the system in the high RH regime as reliable humidity control was possible even beyond the high deliquescence RH of $K_2SO_4$. The humidity values in the individual panels of the figure are uncorrected and represent averaged raw sensor data. The water uptake can be easily recognized from the changes in particle morphology and by the optical density (OD) increase at the oxygen K-edge. (For a definition of the OD,





please refer to Ghorai and Tivanski (2010).) Full deliquescence occurred between 99 and 100 % RH (between Fig. 5b-c), which
agrees well with the aforementioned literature value taking into account the sensor's uncertainty of at least ±3 % RH (specified
only up to 80 % RH by Bosch Sensortec (2019)). The efflorescence happened suddenly during dehydration at 69±5 % RH (see
red arrow in Fig. 5d). In the literature, a value of 60 % RH was reported for the efflorescence relative humidity (DRH) by
Freney et al. (2009). This deviation might be attributed to the presence of the silicon nitride substrate but likely is a result of
the hysteresis of the sensor, which was operated under fairly extreme conditions here and needs a long time to equilibrate after
operation close to saturation conditions. In general, it is therefore recommended to take temperature readings, as well as the
RH values of the two other environmental sensors into consideration when evaluating the humidity inside the microreactor,
instead of relying on just one sensor.

A pressure calibration was not done for the measurements presented here, as the RH is independent from $p$. Furthermore,
the absolute accuracy of ±1.7 hPa, as reported by Bosch Sensortec (2019), was considered sufficiently accurate. However, note
in this context that using helium as a process gas can cause a gradual drift of the measured pressure values, due to helium
permeating the sensor, as was reported by Sparks et al. (2013).

As a second proof-of-performance application, presented in Fig. 6, the hygroscopic growth curve of AS was recorded using
fine-pitched RH steps at the MAXYMUS instrument. Again, the increase of optical density at the O K-edge served as a measure
for the water mass uptake, in analogy to the $NaNO_3$ measurements by Ghorai and Tivanski (2010) and the $(NH_4)_2SO_4$ water
uptake study by Zelenay et al. (2011a). Considering that the data obtained are based on the small number of just four single
AS particles, a good agreement with the AIM model by Clegg et al. (1998), for the deliquescence point and the overall trend
was found.

As a third proof-of-performance application, a water freezing experiment with isoprene SOA particles at high RH was con-
ducted. These results illustrate that the system can be used for controlled freezing experiments to investigate ice nucleation as
well as for kinetic deceleration of fast processes. The particles were produced using a Potential Aerosol Mass (PAM) chamber
(Kang et al., 2007; Lambe et al., 2011) in the presence of AS seed particles to increase the SOA yield, as described by Lambe
et al. (2015), and subsequently impacted on a silicon nitride membrane window. In the course of the experiment, the microre-
actor was cooled from room temperature to about −12 °C with the humidified gas flow fully open to initiate a fast hygroscopic
particle growth. At the desired temperature the gas flow through the microreactor was stopped by closing the two shut-off
valves (compare Fig. 1) to prevent more water from condensing and to prevent ice from nucleating inside the microreactor
body and potentially blocking the gas channel. As the RH reading of sensor S2 quickly went into saturation during the experi-
ment, we can only estimate the relative humidity to be likely above 95 % at the sample location. Subsequently, in-depth X-ray
microspectroscopic analysis at the oxygen K-edge of one exemplary SOA particle revealed an embedded crystalline structure
(Fig. 7a). The recorded hyperspectral NEXAFS data obtained from a subregion of the entire particle were separated by the
NNMA analysis feature of the MANTiS software (Mak et al., 2014) into two main spectral components, which are clearly
localized, either at the matrix or the embedded structure, visualized by the cluster thickness maps (Fig. 7b1 & b2). A compar-
ison of the corresponding spectra (Fig. 7c1 & c2) with reference spectra of liquid water (Sellberg et al., 2014; Nilsson et al.,
2010) and ice (Sellberg et al., 2014; Nilsson et al., 2010; Waldner et al., 2018), suggests that the spectral components represent

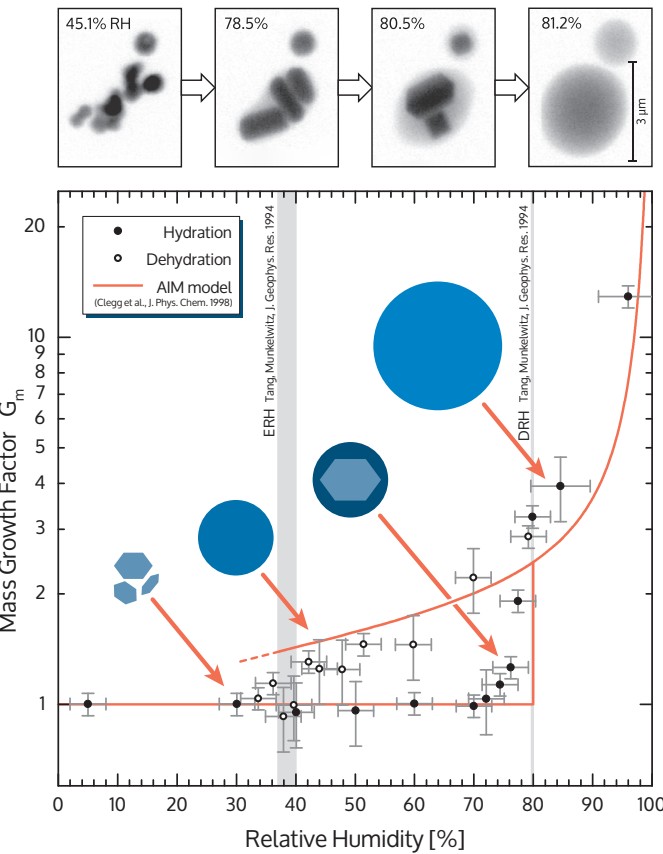

**Figure 6.** Hygroscopic growth curve of individual ammonium sulfate (AS, $(NH_4)_2SO_4$) crystals, based on the oxygen uptake of four lab-prepared micron sized aerosol particles, imaged at the oxygen K-edge. The image sequence at different humidities above, taken from Pöhlker et al. (2014), shows similar AS particles at the oxygen pre-edge (528 eV) and nicely illustrates the water uptake and the Ostwald ripening. The water mass uptake was calculated from OD maps in analogy to the method used by Ghorai and Tivanski (2010) for $NaNO_3$ and by Zelenay et al. (2011a) for $(NH_4)_2SO_4$.



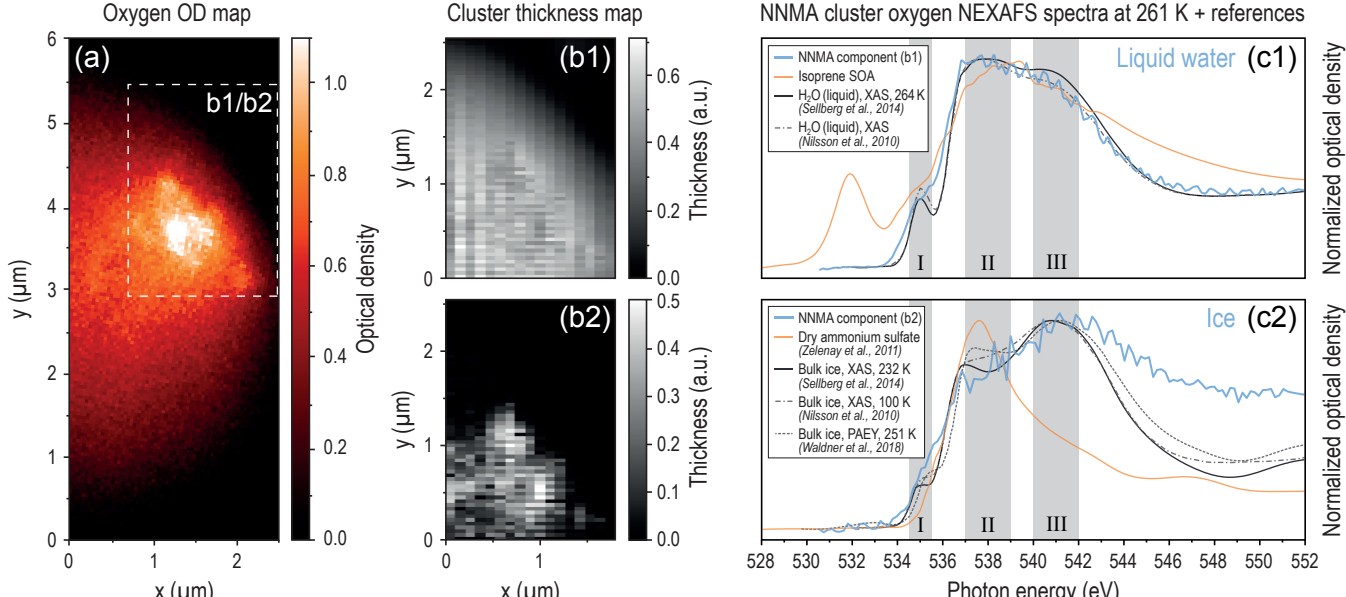

**Figure 7.** Water ice observed at $-12\,°C$ in an aqueous isoprene SOA particle. (a) Optical density map at the oxygen K-edge composed from five single images at different energies: 528.2 and 532.0 eV (pre-edge) and 544.2, 546.2 and 553.2 eV (post-edge). Results from the non-negative matrix approximation (NNMA) analysis are displayed in panels b1, b2, c1, and c2. The cluster thickness maps (weights for hyperspectral pixels) are shown in panels b1 for the matrix component (aqueous phase) and in b2 (ice-enriched phase). Panels c1 and c2 show the corresponding spectra along with reference spectra. Accordingly, the three main spectral features are labeled I, II, III, and emphasized by a grey background in analogy to Bartels-Rausch et al. (2017) and Waldner et al. (2018). All spectra were shifted by 3.2 eV to larger energy values with respect to the reference spectra. The OD map, the thickness maps and the spectral components were extracted from raw data by using the NNMA analysis feature included in MANTiS v.3.0.0.1 (Lerotic et al., 2004, 2005, 2014), commit version #5847171 (Lerotic et al., 2019). (NNMA parameters chosen: k = 4, spectra similarity = 2; spectra smoothness = 0; sparseness = 2.2; number of iterations = 1200; delta error threshold = 0.001)

coexisting water-enriched and ice-enriched phases. Their spectra differ significantly from the spectrum of isoprene SOA under dry conditions (orange spectrum in Fig. 7c1), which was obtained from the same sample prior to wetting, and also does not match the AS reference spectrum (orange spectrum in Fig. 7c2) taken from Zelenay et al. (2011a). We therefore assume that the observed droplet is highly diluted with water and that the characteristic reversal of the intensity ratios of the spectral features II and III (compare blueish spectra in Fig. 7c1 with c2), as a result of a change in coordination geometry and bond strength as

observed by, e.g., Sellberg et al. (2014); Nilsson et al. (2010); and, Waldner et al. (2018), can to a large extent be attributed to $H_2O$ molecules only. The observed ice crystal remained stable in shape over many hours. Further crystal growth was probably kinetically impeded through the viscosity of the matrix solution and/or thermodynamically by the depression of the residual liquid phase's freezing point, which is gradually lowered by precipitation of the ice phase and consequent solute enrichment in the liquid phase, until the freezing stops at a certain composition. This slowing down of the crystal growth was also described





by Budke et al. (2009) for sucrose solutions. Still, it is surprising to find the observed immersion freezing in aqueous isoprene SOA (seeded with AS), as it occurred at an unusually high temperature of $-12\,°\mathrm{C}$, compared to model predictions (e.g., Hoose and Möhler, 2012; Berkemeier et al., 2014). However, other parameters like the humidification rate, for instance, can considerably influence the upper temperature boundary for immersion freezing (Berkemeier et al., 2014). Besides that, the potential presence of ice nucleation-active contaminants, the role of the sample substrate, and an influence of the ionizing X-ray beam
itself can not be excluded.

    In terms of representative statistics of the number of analyzed particles, methods based on STXM-NEXAFS are inherently limited by comparatively long scan times. Accordingly, some of the results on the standard compounds shown here can be obtained more efficiently and probably more precisely with other techniques, such as Hygroscopicity Tandem Differential Mobility Analyzer (HTDMA) systems (Brechtel and Kreidenweis, 2000a, b), and Differential Mobility Analysers coupled to
a Humidified Centrifugal Particle Mass Analyser (DMA-HCPMA) (Vlasenko et al., 2017). Also single particle traps, e.g., the electrodynamic balance (EDB) (Cohen et al., 1987; Tang et al., 1995), and the aerosol optical tweezers are highly accurate, well-established techniques and do not require a substrate (Mitchem and Reid, 2008; Krieger et al., 2012). Therefore, the main purpose of showing these results here is to illustrate the analytical capabilities of the system, particularly at high RH and low $T$.

One of the real analytical strengths of STXM-NEXAFS in combination with MIMiX emerges in the analysis of ambient aerosol particles, since detailed single particle studies can typically not be conducted on site, particularly at remote locations. Thus, sampling of ambient particles onto suitable substrates for a subsequent investigation by offline techniques is required. Such samples are well-suited for in-depth studies with the MIMiX system. Another analytical strength of the system relates to the unique combination of microstructural, hygroscopic, and chemical information, which can be obtained on the level of
individual particles in the submicron particle size range, while being relatively damage-free through the use of soft X-rays in a dose-efficient scanning system.

## 4 Conclusions

This study presents the design, construction, and initial testing of a microreactor system for *in-situ* STXM-NEXAFS analyses of aerosol particles under controlled environmental conditions. Its compact size ensures high portability of the setup, without
sacrificing functionality. The operating ranges cover a wide spectrum of $p$, $T$, and RH conditions, representing large parts of the troposphere. Due to the integrated cooling system, the accessible $T$ and RH are wider than the corresponding ranges in previously reported setups. Moreover, through a compact design of the microreactor, a measurement of the essential environmental parameters very close to the sample has been realized. The microreactor can be operated safely at atmospheric pressure inside the sample chamber, i.e., at a differential pressure of at least $1000\,\mathrm{hPa}$ relative to the microscope chamber. Despite significant
spatial limitations in the STXM optics, the microreactor has been kept compatible to a variety of STXM instruments. For initial measurements, it has been installed at the Helmholtz-Zentrum Berlin (BESSY/MAXYMUS) and at the Swiss Light Source



(SLS/PolLux). The sample exchange mechanism allows quick and convenient substrate changes and minimizes mechanical stress on the fragile samples.

The results from initial experiments (i.e., hygroscopic growth of $(NH_4)_2SO_4$, deliquescence and efflorescense of $K_2SO_4$, as well as the observation of water ice in an aqueous isoprene SOA droplet) confirm that the microreactor is a promising and flexible tool for a variety of *in-situ* particle processing studies in environmental STXM experiments. Particularly, the system allows controlled studies under high RH and/or low $T$ conditions, which are relevant for in-depth investigation of various atmospheric processes.

Future development of the MIMiX system is intended to include an extension of the cooling capabilities for *in-situ* ice nucleation observations and the introduction of an optical fiber to study photochemically driven multiphase reactions. On the software side, integration with the Experimental Physics and Industrial Control System (EPICS) and the Pixelator software is planned in order to store environmental parameters in parallel with the X-ray microscopic data at per pixel resolution.

*Data availability.* The STXM-NEXAFS data used for Fig. 5 and 7, the NNMA analysis results, and the corresponding spectra have been deposited in Edmond, the Max Planck Society's open-access data repository under https://dx.doi.org/10.17617/3.39 (Förster and Pöhlker, 2019). For specific data requests beyond the deposited data, please contact the corresponding authors.

*Video supplement.* A video of the microreactor assembly can be found in the same repository as the scientific data and is available in 720p and 1080p resolution under https://dx.doi.org/10.17617/3.39 (Förster and Pöhlker, 2019).

*Author contributions.* JDF was responsible for the mechanical and electrical design of the microreactor, the conceptual design of the control system and the assembly. CP supervised the construction work. CG and ML developed the microcontroller-based electronic system and helped JDF with the programming of scripts for the graphical user interface. MA, SSS, MW, and BW were consulted in an early design stage and influenced the final design of the microreactor. HT, CP, and JDF prepared the samples. JDF led the writing of the paper. CP, MOA, and UP supervised the paper writing. The adaptation of the microreactor to the STXM instruments were conducted by JDF, CP, and MOA, with the technical assistance of MW, BW, and JR. The measurements were led by JDF, CP, and MOA and supported by FD and PAA. All authors contributed to the paper finalization.

*Competing interests.* There are no conflicts to declare.

*Acknowledgements.* This work was supported by the Max Planck Society (MPG). The authors thank Thomas Kennter, Frank Kunz, and the MPIC's mechanical workshop team for their excellent work. We acknowledge the Helmholtz-Zentrum Berlin, Germany for the alloca-



tion of the synchrotron radiation beam time at BESSY II and the Paul Scherrer Institute, Villigen, Switzerland for provision of synchrotron radiation beamtime at the PolLux beamline of the SLS. The PolLux endstation was financed by the Federal Ministry of Education and Research (BMBF) through contracts 05KS4WE1/6 and 05KS7WE1. We thank Michael Bechtel, and Blagoj Sarafimov for technical assistance during the beamtimes. We further thank Frank Helleis, Ralf Wittkowski, Mario Birrer, Thomas Berkemeier, Stefan Blanckart, and Berthold Kreuzburg for their support and stimulating discussions.





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
