# Peer review of "MIMiX: A Multipurpose In-situ Microreactor system for X-ray microspectroscopy to mimic atmospheric aerosol processing"

_Atmospheric Measurement Techniques, 2019_

## Referee Comment (RC1) · Anonymous Referee #1 · 22 Feb 2020

General comments.

The manuscript presents instrumental details and some demonstrations of a microreactor that can fit inside a STXM/NEXAFS apparatus to enable X-ray microscpectroscopy studies under a wider range of environmental conditions than are usually present for STXM. The ability to control temperature is a very nice addition to this area of the field. This is a very clear manuscript that communicates the capabilities of the technique well. However, I recommend the addition of details to the text that provide context for some of the time-scales and other limitations of these experiments. I recommend this for publication in AMT after the following minor comments are addressed.

Minor comments

1. Emphasis is placed on the low profile in the front enabling C K-edge analysis. Is there sufficient room to enable S K-edge as well? If not, what aspect is limiting this?

2. It is noted that "We could not detect any carbonaceous contaminants from potential outgassing of the O-rings, from the lubricant or from the glued components". Under what systems was this probed? Was it only for the systems shown in the manuscript or have a broader range been tested? I am specifically concerned about the glue and any fumes that could end up partitioning into organic aerosol particles.

3. On page 11 you state: "This deviation might be attributed to the presence of the silicon nitride substrate but likely is a result of the hysteresis of the sensor, which was operated under fairly extreme conditions here and needs a long time to equilibrate after operation close to saturation conditions". How long is a "long time"? What are the typical time scales for these experiments and what else (if anything) limits the time scales (aside from the length of time it takes to raster scan and collect the spectrum)?

4. For the isoprene experiment, how quickly was the system cooled? How much variation in this cooling rate do you have? What about for warming rates? Is there any equilibration time needed between cooling and warming cycles?

5. For the isoprene study did any other particles nucleate ice? How wide of a field of view is possible with this apparatus? Can you still do larger scan areas to enable a search for "exemplary" particles? Are there any limitations to this in the current design?

6. How long is a typical experiment for both RH only and the ice nucleation studies (T and RH varying)?

7. Some of the font sizes in the figures are rather small and difficult to read. I particularly recommend improvements in Figure 1 and the axis on Figure 4 (the pale colors are very hard to read when printed in black and white).

---

## Referee Comment (RC2) · Anonymous Referee #2 · 31 Mar 2020

This work reports a new microreactor system for X-ray microscopy, which facilitates observations under in-situ conditions and extends the accessible parameter ranges of previously reported setups to very high humidities and low temperatures. The authors present the design and analytical scope of the system, along with results from hydration experiments on ammonium sulfate and the observation of water ice at low temperature and high relative humidity in a secondary organic aerosol particle from isoprene oxidation. The work is of scientific significance, and conclusions well supported by the data. It can be accepted after addressing the following issues: 1. What is the time resolution and space resolution for the measurements. 2. The authors mentioned that "We could not detect any carbonaceous contaminants from potential outgassing

of the O-rings, from the lubricant or from the glued components", more evidences are needed for this claim. 3. Is it possible to perform in-situ heterogeneous reaction study using this equipment?

---

## Author Response (AR1)

**Revision of amt-2019-507**

MIMiX: A Multipurpose In-situ Microreactor system for X-ray microspectroscopy to mimic atmospheric aerosol processing

by J.-D. Förster *et al.*

This document includes:

June 9, 2020

**I. Point-by-point response to Anonymous Referee #1**

We appreciate the very positive feedback and the inquisitive questions by Referee #1, which helped us to improve the quality of our manuscript. The referee's comments and our responses are detailed below.

1. Referee Comment: Emphasis is placed on the low profile in the front enabling C K-edge analysis. Is there sufficient room to enable S K-edge as well? If not, what aspect is limiting this?

Author response: With the current setup the sulfur edge cannot be reached. However it would be possible with only minor modifications. To our knowledge only STXM beamlines 11.0.2 at the Advanced Light Source (Berkeley, CA, USA) and UE46 at BESSY II (Berlin, Germany) are currently capable of providing photons with energies below 250 eV and high quality imaging at such a low photon energy is still challenging due to the shallow depth of field and the high demands on the quality of suitable x-ray optics. The corresponding paragraph on p.5–6 was changed accordingly:

> With a typical focal length of 1.36 mm at 280 eV, the space between the focal plane and the zone plate (compare Fig. 3a and Fig. 1 in Huthwelker et al. (2010)), with the order sorting aperture (OSA) in between, is very limited. The OSA (60 μm
> 120   diameter) is located approximately 320 μm away from the sample plane to prevent unfocused X-rays and stray light from passing through the sample and from entering the detector. Subtracting the silicon frame thickness of the front window leaves just 120 μm between the OSA and the microreactor. We therefore avoided a sample holder design that adds more material to this side of the microreactor. Due to inherent geometric restrictions it is not possible to reach energies below 200 eV with the current setup. However, silicon nitride windows with just 100 μm thick frames are commonly available and their use together
> 125   with a modified mounting disk would in principle bring the sulfur L-edge at 170 eV into range, neglecting any limitation by X-ray optics and insertion devices. It has been shown that it is feasible to image the sulfur distribution in aerosol particles at the STXM beamline 11.0.2 at the Advanced Light Source (Berkeley, CA, USA) (Hopkins et al., 2008) and at MAXYMUS (Pöhlker et al., 2014, Fig. S6).

2. Referee Comment: It is noted that "We could not detect any carbonaceous contaminants from potential outgassing of the O-rings, from the lubricant or from the glued components". Under what systems was this probed? Was it only for the systems shown in the manuscript or have a broader range been tested? I am specifically concerned about the glue and any fumes that could end up partitioning into organic aerosol particles.

Author response: In initial tests the "KaWeS - Joint grease without silicon" from J.P. Pöllath - Labor-Technologie was used to lubricate the O-rings. Its vapor pressure is $10^{-6}$ hPa vs. the lower vapor pressure $10^{-9}$ hPa for Apiezon N. The figure below was added to the supplement to illustrate how contamination appears under the microscope. It is clearly visible how carbonaceous material is dissolved in the aqueous phase and additionally accumulates in a ring around the particle.

[Figure]

**Figure S3:** Carbonaceous contamination on ammonium sulfate particles at 65% RH using "KaWeS - Joint grease without silicon" as lubricant. a) Carboxylate map from energies 282.0 and 288.5 eV; b) Nitrogen map from energies 396.0 and 406.0 eV; c) Oxygen map from energies 527.0 and 537.0 eV

The corresponding paragraph on p.8 was rephrased to give more background information on our development process and to convince the reader of our awareness of the issue.

> gas stream through an O-ring-sealed recess in the metal body (Fig. 3B). A total number of three O-rings seal the microreactor from the microscope. The O-rings were slightly lubricated with Apiezon N Cryogenic High Vacuum Grease to assure a tight seal and a smooth rotation of the sample locking mechanism.  This solved the initial problem we had with a different
> 150 high vacuum grease, which introduced an organic contamination into the samples (Fig. S3). We regularly conduct carbon K-edge spectroscopy to identify beam damage and potential sources of contamination in our analysis, but do not detect any  impurities originating from microreactor components, even in studies where particles were processed over many hours (e.g. Alpert et al., 2019) . We attribute this to the fact that the microreactor is constantly flushed with fresh
> 155 process gas at comparably high flow rates, which keeps the amount of impurities originating from the lubricant, O-rings, or glued components at a low concentration or at least below our detection ability using STXM-NEXAFS spectroscopy.

3. Referee Comment: On page 11 you state: "This deviation might be attributed to the presence of the silicon nitride substrate but likely is a result of the hysteresis of the sensor, which was operated under fairly extreme conditions here and needs a long time to equilibrate after operation close to saturation conditions". How long is a "long time"? What are the typical time scales for these experiments and what else (if anything) limits the time scales (aside from the length of time it takes to raster scan and collect the spectrum)?

Author response: In our experiments we usually allowed the RH to stabilize for a few minutes before we recorded images or spectra. In Fig. 4a the response characteristics at approximately 80 % RH are shown, where, after each step, the system became sufficiently stable again after 2 minutes at maximum. The response time is almost independent of the increment. Only at higher relative humidities, i.e., close to saturation conditions, the equilibration takes longer due to surface ad- and absorption of water molecules but never exceeded ten minutes. Accordingly, we rephrased "a long time" to "up to ten minutes" on p.10, l.213. A good example of a common experiment is given by the description of Fig. 6 in lines 221–232 where we added more information about the duration of the overall hydration/dehydration experiment.

> As a second proof-of-performance application, presented in Fig. 6, the hygroscopic growth curve of AS was recorded using fine-pitched RH steps at the MAXYMUS instrument. The figure contains data from two independent measurements on four AS particles each. Again, the increase of optical density at the  oxygen K-edge served as a measure for the water  uptake, in analogy to the $NaNO_3$ measurements by Ghorai and Tivanski (2010) and the $(NH_4)_2SO_4$ water uptake
> 225 study by Zelenay et al. (2011a).  A good agreement with the AIM model  II by Clegg et al. (1998) for the deliquescence point and the overall trend was found. In terms of representative statistics of the number of analyzed particles, methods based on STXM-NEXAFS are inherently limited by comparatively long scan times. As a rough measurement time estimate, the recording of a full hydration/dehydration cycle with 22 RH steps took two hours in case of dataset 1. The scan time itself was
> 230 about two minutes at each RH step for recording images at two different energies (65 nm pixel size, 1 ms dwell time per pixel on a $154 \times 122 \, px^2$ area). The remaining time was spent on waiting for the RH to stabilize, which is particularly important during dehydration experiments and at high relative humidities, as was mentioned above.

In addition, we reanalyzed the data for the corresponding Fig. 6. We now integrated the single particles visible in the field of view instead of taking the average across all particles and we added a second dataset to give more statistical weight to our analysis and to demonstrate that a hygroscopic growth curve with a reasonable amount of data points can be recorded in a reasonable amount of time with this technique. Moreover, we found an error in our AIM model calculation, which resulted in an incorrect model curve. Since the method we used here only gives us a measure of the oxygen mass, the model now includes only the oxygen containing species, resulting in an Oxygen Mass Growth Factor instead of a total Mass Growth Factor. The corresponding raw data and calculations have been added to the Edmond repository mentioned in the data availability section. The new figure is attached below:

[Figure]

Figure 6: Hygroscopic growth curve of  ammonium sulfate (AS, $(NH_4)_2SO_4$)  based on  four  AS particles  each,  ranging from 500 nm – 2 µm in size. The image sequence above at different humidities, taken from Pöhlker et al. (2014), shows similar AS particles at the oxygen pre-edge (528 eV) and  illustrates the water uptake and the Ostwald ripening. The  oxygen mass  uptakes of individual particles were calculated from OD maps  (527/560 eV) at the oxygen K-edge. All data were normalized to the  dry particles (encircled data points). Only masses from oxygen containing species contributed to the AIM model II curves (Clegg et al., 1998) . Error bars represent the sensor accuracy (±3% RH below and  ±5% RH above 80% RH) and an estimated measurement error (±0.2 oxygen mass growth units).

Besides that, as mentioned by the referee, only scan times and, in addition, the times for tuning the instrument (e.g., focusing, selection of energies, search for a region of interest on the sample) limit the data aquisition speed.

4. Referee Comment: For the isoprene experiment, how quickly was the system cooled? How much variation in this cooling rate do you have? What about for warming rates? Is there any equilibration time needed between cooling and warming cycles?

Author response: The cooling rate was not constant. The average cooling rate between 289 K and 273 K was about 1 K min$^{-1}$ and the target temperature of 261 K was reached after one hour. Due to the setting of the PI regulation the temperature approached the setpoint asymptotically in this experiment. No equilibration time is needed between cooling and warming cycles as the circulation pump can be started, stopped and varied in speed immediately. However, the Stirling cooler needs an initial startup time of about 30 minutes before cooling experiments can be conducted due to the mass of the cooling head and its corresponding heat capacity. The cooling and warming rates are approximately equivalent, but the warming rate depends on the quality of the surrounding vaccum. To better quantify cooling and warming rates we rephrased p.14, l.244–248

245  by Lambe et al. (2015), and subsequently impacted on a silicon nitride membrane window. In the course of the experiment,  starting from 289 K, the target temperature of 261 K was reached within an hour with an initial average cooling rate of about 1 K min$^{-1}$ between 289 K and 273 K and with the humidified gas flow fully  enabled to initiate a fast hygroscopic particle growth. The maximum cooling rate achieved with this system was 2.5 K min$^{-1}$.

and p.14, l.269–271, respectively.

270  _even though the temperature fluctuated by ±0.2 K. During warming up from 261 K to 269 K at a rate of 2.7 K min⁻¹ a restructuring of the crystalline structure could be observed. (The warming rate largely depends on the quality of the surrounding vacuum since no active heating is involved.)_

5. Referee Comment: For the isoprene study did any other particles nucleate ice? How wide of a field of view is possible with this apparatus? Can you still do larger scan areas to enable a search for "exemplary" particles? Are there any limitations to this in the current design?

Author response: Unfortunately, we had no time to study more particles during this experiment, but we would like to investigate this further in follow-up studies. Since data are sparse, we rephrased the corresponding paragraph on p.14–15, l.264–279:

be attributed to $H_2O$ molecules only. The  _crystalline structure could be observed over many hours and_
265 remained stable in shape ~~over many hours. Further crystal growth was probably kinetically impeded through the viscosity of the matrix solution and/or thermodynamically by the depression of the residual liquid phase's freezing point, which is gradually lowered by precipitation of the ice phase and consequent solute enrichment in the liquid phase, until the freezing stops at a certain composition. This slowing down of the crystal growth was also described by Budke et al. (2009) for sucrose solutions. Still, it is surprising to find the observed~~ _even though the temperature fluctuated by ±0.2 K. During warming up_
270 _from 261 K to 269 K at a rate of 2.7 K min⁻¹ a restructuring of the crystalline structure could be observed. (The warming rate largely depends on the quality of the surrounding vacuum since no active heating is involved.) It should be noted that it is unlikely to find_ immersion freezing in aqueous isoprene SOA (seeded with AS), as it occurred at an unusually high temperature of  _261_ K, compared to model predictions (e.g., Hoose and Möhler, 2012; Berkemeier et al., 2014). However, other parameters like the humidification rate, for instance, can considerably influence the upper temperature boundary for
275 immersion freezing (Berkemeier et al., 2014). Besides that, the  presence of ice nucleation-active contaminants, the role of the sample substrate, and an influence of the ionizing X-ray beam itself can not be excluded. _Due to the loss of beam and the subsequent end of the beamtime, the melting process could not be followed till the end. Although the data are sparse for giving actual proof for ice formation, we found our observation worth reporting and feel encouraged to conduct further investigations on this in follow-up studies._

In general, there is no limitation of the field of view. The whole sample window could be scanned. However, the fine stage, which allows a more efficient data aquistion due to fast piezo motors is limited to movements of about 30 micrometers (in case of the MAXYMUS instrument). A common mode of operation is to 'jump' to a new location with the coarse stage and record scans with the fine stage in the newly selected area. Our microreactor design does not limit movements of the stages as far as required for scanning the sample window. In Fig. 3c the 1 mm clearance around the detector tip is shown, which is the hard limit for the coarse stage movement. Since the technical limitations like scan ranges of the instruments are generally not influenced by our microreactor and are very instrument specific, we would like refer to the mentioned literature for MAXYMUS (Follath et al., 2010; Nolle et al., 2011; Weigand, 2014) and PolLuX (Flechsig et al., 2007; Raabe et al., 2008; Frommherz et al., 2010).

6. Referee Comment: How long is a typical experiment for both RH only and the ice nucleation studies (T and RH varying)?

Author response: We hope that our answers to questions 3-5 are sufficiently detailed to give a rough estimate of the time consumption of such experiments. The cooling system control is independent of the RH control system. The cooling system can be driven with a cooling/warming rate of about 2.5 K min$^{-1}$ as was mentioned above, while the humidity can be increased or lowered much faster.
We are stating on p.8, l.166–169:

166                      Steep $p$ and RH gradients (e.g., from
 dry conditions to above 80 % RH within less than a minute) can be achieved in a reliable and reproducible manner. Besides a
168 stepwise increase of the humidity, it is possible to run preprogrammed ramps or periodically repeated hydration/dehydration
 cycles to mimic dynamic atmospheric processes.

and on p.9, l.178–180:

178 avoid undesired water condensation inside the microreactor. Therefore, the $T$ control system was designed for high-precision
 regulation and not for high cooling (~ 2.5 K min$^{-1}$) or warming rates. In Fig. 4b, the high stability of the regulator set to a
180 target $T$ of 283.2 K is shown over a 4-hour time span.

7. Referee Comment: Some of the font sizes in the figures are rather small and difficult to read. I particularly recommend improvements in Figure 1 and the axis on Figure 4 (the pale colors are very hard to read when printed in black and white).

Author response: Thank you for pointing this out. Figures 1 and 4 have been replaced, the font sizes as well as the saturation of the colors has been increased. The new figures are shown below:

[Figure]

Figure 1: Gas Flow and cooling system schematics of the entire system. Left: X-ray microscope chamber with sketched microreactor. Refer to to Fig. 2 for a detailed view of the microreactor. Right: 19-inch enclosure that includes the control system. S1-S4: p, T, RH environmental sensors (Bosch BME280); M: Solenoid operated shut off valves (Bronkhorst® EV-02-NC-V); TC: Thermocouple (Omega™ 5TC-TT-TI-4)

[Figure]

Figure 4: a) Response characteristic of the humidifier at different setpoints. blue: humidity measured at S1 (control variable); orange: humidity measured at S2 (compare with compare with with Fig. 1) at an ambient temperature between 301.2–302.2 K and a microreactor $T$ of 283.2 K. b) Microreactor $T$ stability over ~240 min at a setpoint of 283.2 K. The gray shading emphasizes a ±0.025 K margin.

**II. Point-by-point response to Anonymous Referee #2**

We appreciate the short and helpful comments by referee #2, which have been considered carefully and helped to improve the quality of our manuscript. The referees' comments and our responses are outlined below.

1. Referee Comment: What is the time resolution and space resolution for the measurements?

Author response: The answer to this question highly depends on the required resolution in space and time. It also depends on the desired amount of pixel statistics and energy resolution. In the description of our newly composed Fig. 6 we provided a rough time estimate by stating on p.11–13, l.227–232:

> point and the overall trend was found. In terms of representative statistics of the number of analyzed particles, methods based on STXM-NEXAFS are inherently limited by comparatively long scan times. As a rough measurement time estimate, the recording of a full hydration/dehydration cycle with 22 RH steps took two hours in case of dataset 1. The scan time itself was
> 230 about two minutes at each RH step for recording images at two different energies (65 nm pixel size, 1 ms dwell time per pixel on a $154 \times 122$ px$^2$ area). The remaining time was spent on waiting for the RH to stabilize, which is particularly important during dehydration experiments and at high relative humidities, as was mentioned above.

Our microreactor does not alter the performance and resolution of STXM instruments. In the literature mentioned in the manuscript the achievable spatial resolution reported varies between 15-25 nm for MAXYMUS (Follath et al., 2010; Nolle et al., 2011; Weigand, 2014) and PolLuX (Raabe et al., 2008; Frommherz et al., 2010).

2. Referee Comment: The authors mentioned that "We could not detect any carbonaceous contaminants from potential outgassing of the O-rings, from the lubricant or from the glued components", more evidences are needed for this claim.

Author response: Indeed we detected carbonaceous contaminants in very initial tests. We analyzed the problem thoroughly and could trace it back to the lubricant we first used. After replacing the lubricant, no contamination was visible to us in subsequent experiments. A clear sign of carbonaceous contamination is shown in Fig. S3, which we added to the supplement. On p.8 we provided additional information to address our learning curve with this issue:

> gas stream through an O-ring-sealed recess in the metal body (Fig. 3B). A total number of three O-rings seal the microreactor from the microscope. The O-rings were slightly lubricated with Apiezon N Cryogenic High Vacuum Grease to assure a tight seal and a smooth rotation of the sample locking mechanism.  This solved the initial problem we had with a different
> 150 high vacuum grease, which introduced an organic contamination into the samples (Fig. S3). We regularly conduct carbon K-edge spectroscopy to identify beam damage and potential sources of contamination in our analysis, but could not detect any  impurities originating from microreactor components, even in studies where particles were processed over many hours (e.g. Alpert et al., 2019). We attribute this to the fact that the microreactor is constantly flushed with fresh
> 155 process gas at comparably high flow rates, which keeps the amount of impurities originating from the lubricant, O-rings, or glued components at a low concentration or at least below our detection ability using STXM-NEXAFS spectroscopy.

Please also refer to our answer to question #2 of Referee #1 for additional information.

3. Referee Comment: Is it possible to perform in-situ heterogeneous reaction study using this equipment?

Author response: The in-depth study of diffusion limited reactions relevant in atmospheric multiphase chemistry was one motivation to build this setup. We already observed the diffusion limited oxidation of $Fe^{2+}$ in aerosol particles composed of xanthan gum and $FeCl_2$ and are planning to conduct more experiments under atmospherically relevant pressure, temperature, and humidity regimes. Accordingly, we reorganized the paragraph on p.15 l.280–292 and added the study (Alpert et al., 2019) as a reference:

[revised manuscript text omitted]